# Insertion-sequence-mediated mutations both promote and constrain evolvability during a long-term experiment with bacteria

Jessika Consuegra [1,7,8], Joël Gaffé[1,8], Richard E. Lenski[2,3], Thomas Hindré[1], Jeffrey E. Barrick [3,4], Olivier Tenaillon[5,6] & Dominique Schneider [1✉]

Insertion sequences (IS) are ubiquitous bacterial mobile genetic elements, and the mutations they cause can be deleterious, neutral, or beneficial. The long-term dynamics of IS elements and their effects on bacteria are poorly understood, including whether they are primarily genomic parasites or important drivers of adaptation by natural selection. Here, we investigate the dynamics of IS elements and their contribution to genomic evolution and fitness during a long-term experiment with *Escherichia coli*. IS elements account for ~35% of the mutations that reached high frequency through 50,000 generations in those populations that retained the ancestral point-mutation rate. In mutator populations, IS-mediated mutations are only half as frequent in absolute numbers. In one population, an exceptionally high ~8-fold increase in IS*150* copy number is associated with the beneficial effects of early insertion mutations; however, this expansion later slowed down owing to reduced IS*150* activity. This population also achieves the lowest fitness, suggesting that some avenues for further adaptation are precluded by the IS*150*-mediated mutations. More generally, across all populations, we find that higher IS activity becomes detrimental to adaptation over evolutionary time. Therefore, IS-mediated mutations can both promote and constrain evolvability.

[1] Univ. Grenoble Alpes, CNRS, Grenoble INP, TIMC-IMAG, Grenoble, France. [2] Department of Microbiology and Molecular Genetics, Michigan State University, East Lansing, MI, USA. [3] BEACON Center for the Study of Evolution in Action, Michigan State University, East Lansing, MI, USA. [4] Department of Molecular Biosciences, The University of Texas at Austin, Austin, TX, USA. [5] IAME, UMR 1137, INSERM, Paris, France. [6] IAME, UMR 1137, Université Paris Diderot, Sorbonne Paris Cité, Paris, France. [7] Present address: Institut de Génomique Fonctionnelle de Lyon (IGFL), Ecole Normale Supérieure de Lyon, CNRS, UCBL1 Université de Lyon UMR 5242, Lyon, France. [8] These authors contributed equally: Jessika Consuegra, Joël Gaffé. ✉email: dominique.schneider@univ-grenoble-alpes.fr

Adaptation by natural selection requires genetic variation and, in turn, mutations that produce that variation. However, a higher proportion of mutations are deleterious than are beneficial, and so the molecular pathways that control the production of diversity reflect a dynamic tension between evolvability and stability over both short and long time scales[1,2]. This tension has been studied in the context of the evolution of point-mutation rates and gene-regulatory networks. However, it is not well understood with respect to the dynamics of mobile genetic elements, such as Insertion Sequence (IS) families in bacteria.

Both theory and experiments have shown that mutation rates can evolve in a dynamic manner. Hypermutator strains of bacteria, which have defects in DNA metabolism and repair genes, have been observed in many clinical settings and laboratory experiments[3–5]. By producing more beneficial mutations, hypermutators can increase in frequency in populations that are adapting to new or constantly changing environments[6]. However, hypermutability also increases a population's load of deleterious mutations. Hence, after the rise of a hypermutator, selection often favors reversions or compensatory mutations that reduce the mutation rate, especially after a population becomes well adapted to its new environment[2,7].

IS elements are small (~0.7 to ~2.5 kbp), mobile genetic elements found in most bacterial genomes. There is tremendous variability in the types and copy numbers of the different IS element families that bacteria harbor[8]. Both their presence and activities affect genome structure and gene expression, and they can thus impact fitness[9,10].

From an evolutionary perspective, IS elements can have opposing effects on fitness, which has led to debate about how they persist in bacterial populations[8–11]. On the one hand, many or most transpositions of IS elements are thought to reduce fitness. These effects are harmful not only for the host cell, but also for the elements. Lineages of cells that carry them are more likely to be removed by selection because they will tend to produce more deleterious mutations. To counter their losses due to selection, IS elements are often cast as genomic parasites that can be maintained in populations only by high rates of transposition and horizontal transfer[11]. Accordingly, transposition should be tightly regulated by both the element itself and host factors, so as to minimize harmful mutations while still allowing for movement onto and horizontal transfer by conjugative and viral agents[9]. On the other hand, IS elements can also generate beneficial mutations. In laboratory experiments, IS elements have been shown to produce many fitness-enhancing mutations under diverse environmental conditions[12–15]. Moreover, they can contribute to reductive genome evolution in nature by causing gene inactivation and chromosomal deletions that are beneficial when free-living forms transition to host-restricted states[16]. Thus, the beneficial effects of some IS-mediated mutations may be sufficient to sustain, or at least contribute to, IS element persistence[9,14].

To examine the tension between the beneficial and harmful effects of IS elements, we followed the dynamics of mutations caused by IS elements in experimental populations evolving under controlled conditions. In 1988, 12 Escherichia coli populations were founded from a common ancestor, and they have evolved for over 60,000 generations in an environment with glucose as the limiting resource[13,17]. The IS elements in two populations were previously analyzed after 10,000 generations, revealing a dramatic, but unexplained, increase in the number of IS150 copies in one of them[14]. This increase was confirmed in a genomic analysis of 264 evolved clones sampled through 50,000 generations from the 12 populations[18]. Here, we investigate the dynamics of the distribution, regulation, and overall contribution to genome evolution of the IS elements in the 12 populations

during those 50,000 generations. We use genomic and metagenomic data from this experiment[17,18] to correlate the dynamics of IS elements with the fitness improvements of each population, and also to study the interplay between changes in the rates of IS-related mutations and point mutations. In this work, we show that IS-mediated mutations accumulated at different rates in the long-term evolution experiment (LTEE) with E. coli, depending on whether or not a lineage had evolved point-mutation hypermutability. We further show that IS activity tended to promote adaptation early in the LTEE, but it became detrimental over time. Thus, IS-mediated changes can both promote and constrain evolvability.

## Results

We investigated the dynamics of IS elements in the LTEE with E. coli using two sources of information: the whole genomes of individual clones sampled at 11 time points[18] and metagenomes from entire populations sampled at 500-generation intervals[17]. Individual genomes provide a precise accounting of how many mutations accumulated in a lineage leading to a specific cell, including those that were present at low frequencies at one time but then later disappeared, perhaps by being outcompeted. Metagenomes find alleles that reached a detection level of ~ 5–10% in the population. Both sources also allow us to identify the most successful mutations that eventually became fixed in each evolving population (i.e., reached 100% frequency).

**Genomic distribution of IS elements after 50,000 generations.** E. coli REL606 is the ancestor of the LTEE, and its genome harbors 12 distinct types of IS elements, either as one or more complete (IS1, IS2, IS3, IS4, IS30, IS150, IS186, IS600, IS911) or incomplete (ISEc1, ISECB1, ISEhe3) copies[19]. We first compared the genomes of 264 derived clones sequenced from the 12 populations (2 clones from each population at 11 time points through 50,000 generations)[18] to the ancestral genome to quantify the cumulative changes in copy number and genome distribution of all 12 IS elements over evolutionary time (Table 1 and Supplementary Data 1).

As previously observed[14,18], IS150 was the most active element with 9.4 new insertions, on average, in the 50,000-generation clones sampled from the 12 populations. With five ancestral copies, that number implies an average of 1.9 new insertions per ancestral copy. By contrast, IS186 and IS3 had the same ancestral copy number as IS150, but both of them were much less active, with an average of 0.46 and 0.07 new insertions per ancestral copy, respectively. Three other IS elements (IS2, IS4, IS30) were present as single copies in the ancestor. Both IS2 and IS30 were completely inactive during the LTEE, while IS4 generated an average of 0.38 new insertions. The IS1 element, with 28 ancestral copies, had an average of 2.63 new insertions in the 50,000-generation clones, but with the high initial number that corresponds to an average of only ~0.09 new insertions per ancestral copy. Overall, the ancestral copy numbers of the IS element families do not predict their activity in the LTEE.

IS element activity also varied greatly across the 12 populations. Four populations (Ara−4, Ara+3, Ara+4, and Ara+6) had just 2 to 7 new insertions in the 50,000-generation clones across all IS families (Table 1). In striking contrast, the 50,000-generation clones from populations Ara+1 and Ara−3 had between 29 and 41 new insertions. Most of this variation reflects differences in IS150, the most active element, which had a range of 0 to 35 new insertions, with the extremes observed in Ara−4 and Ara+1, respectively (Table 1 and Supplementary Data 1). This extreme variation implies that the evolutionary dynamics of the IS elements must have depended strongly on the specific

**Table 1 Copy number of IS elements in the genomes of the ancestor and evolved clones sampled after 50,000 generations of the LTEE.**

| Population | Clone | IS1 | IS2[a] | IS3 | IS4 | IS30 | IS150 | IS186 | IS600[a] | IS911[a] | New insertions |
|---|---|---|---|---|---|---|---|---|---|---|---|
| Ancestor | REL606 | 28 | 1 | 5 | 1 | 1 | 5 | 5 | 1 | 2 | |
| Ara+1 | 11392 | 30 (2; 0) | – | 4 (0; 1) | 1 (0; 0) | 1 (0; 0) | 40 (35; 0) | 6 (1; 0) | – | – | 38 |
| Ara+1 | 11393 | 32 (4; 0) | – | 4 (0; 1) | 1 (0; 0) | 1 (0; 0) | 40 (35; 0) | 8 (3; 0) | – | – | 41[b] |
| Ara+2 | 11342 | 31 (4; 1) | – | 5 (0; 0) | 1 (0; 0) | 1 (0; 0) | 8 (3; 0) | 9 (4; 0) | – | – | 10[b] |
| Ara+2 | 11343 | 31 (4; 1) | – | 5 (0; 0) | 1 (0; 0) | 1 (0; 0) | 8 (3; 0) | 9 (4; 0) | – | – | 10[b] |
| Ara+3 | 10953 | 28 (0; 0) | – | 4 (0; 1) | 1 (0; 0) | 1 (0; 0) | 5 (0; 0) | 7 (2; 0) | – | – | 3 |
| Ara+3 | 10954 | 30 (2; 0) | – | 5 (0; 0) | 1 (0; 0) | 1 (0; 0) | 5 (0; 0) | 7 (2; 0) | – | – | 5 |
| Ara+4 | 11348 | 28 (1; 1) | – | 5 (0; 0) | 3 (2; 0) | 0 (0; 1) | 6 (1; 0) | 5 (0; 0) | – | – | 5 |
| Ara+4 | 11349 | 29 (2; 1) | – | 5 (0; 0) | 3 (2; 0) | 1 (0; 0) | 7 (2; 0) | 6 (1; 0) | – | – | 7 |
| Ara+5 | 11367 | 31 (3; 0) | – | 5 (0; 0) | 1 (0; 0) | 1 (0; 0) | 15 (10; 0) | 8 (3; 0) | – | – | 16 |
| Ara+5 | 11368 | 31 (3; 0) | – | 5 (0; 0) | 1 (0; 0) | 1 (0; 0) | 15 (10; 0) | 8 (3; 0) | – | – | 16 |
| Ara+6 | 11370 | 27 (0; 1) | – | 5 (0; 0) | 1 (0; 0) | 1 (0; 0) | 6 (1; 0) | 5 (1; 1) | – | – | 2 |
| Ara+6 | 11371 | 32 (4; 0) | – | 5 (0; 0) | 1 (0; 0) | 1 (0; 0) | 6 (1; 0) | 6 (1; 0) | – | – | 6 |
| Ara−1 | 11330 | 30 (2; 0) | – | 5 (0; 0) | 1 (0; 0) | 1 (0; 0) | 12 (7; 0) | 7 (2; 0) | – | – | 11 |
| Ara−1 | 11331 | 30 (2; 0) | – | 5 (0; 0) | 1 (0; 0) | 1 (0; 0) | 12 (7; 0) | 7 (2; 0) | – | – | 11 |
| Ara−2 | 11335 S | 29 (2; 1) | – | 8 (3; 0) | 1 (0; 0) | 1 (0; 0) | 21 (16; 0) | 7 (2; 0) | – | – | 23 |
| Ara−2 | 11333 L | 31 (5; 2) | – | 5 (0; 0) | 2 (1; 0) | 1 (0; 0) | 5 (0; 0) | 4 (0; 1) | – | – | 6 |
| Ara−3 | 11364 | 25 (3; 6) | – | 3 (1; 3) | 0 (0; 1) | 1 (0; 0) | 25 (21; 1) | 12 (7; 0) | – | – | 30[b] |
| Ara−3 | 11365 | 25 (2; 5) | – | 3 (1; 3) | 1 (0; 0) | 1 (0; 0) | 25 (21; 1) | 12 (7; 0) | – | – | 29[b] |
| Ara−4 | 11336 | 26 (1; 3) | – | 4 (0; 1) | 1 (0; 0) | 1 (0; 0) | 5 (0; 0) | 5 (1; 1) | – | – | 2 |
| Ara−4 | 11337 | 27 (2; 3) | – | 4 (0; 1) | 1 (0; 0) | 1 (0; 0) | 5 (0; 0) | 5 (1; 1) | – | – | 3 |
| Ara−5 | 11339 | 31 (4; 1) | – | 4 (0; 1) | 1 (0; 0) | 1 (0; 0) | 19 (14; 0) | 10 (5; 0) | – | – | 23 |
| Ara−5 | 11340 | 31 (4; 1) | – | 4 (0; 1) | 2 (1; 0) | 1 (0; 0) | 19 (14; 0) | 10 (5; 0) | – | – | 24 |
| Ara−6 | 11389 | 30 (2; 0) | – | 7 (2; 0) | 2 (1; 0) | 1 (0;0) | 23 (18; 0) | 7 (2; 0) | – | – | 25 |
| Ara−6 | 11390 | 33 (5; 0) | – | 6 (1; 0) | 2 (1; 0) | 1 (0; 0) | 12 (7; 0) | 8 (3; 0) | – | – | 17 |
| New insertions | | 63 | 0 | 8 | 9 | 0 | 228 | 55[b] | 0 | 0 | |

The numbers of IS gains and losses, respectively, compared to the ancestor are indicated in parentheses for each clone.
[a]No changes were detected in any of the 50,000-generation clones for these three IS elements.
[b]The total number of new IS insertions differs from the sum of IS element gains because chromosomal duplications, including a region containing an IS186 element, resulted in additional copies (2 in both Ara+2 clones, 1 in both Ara−2 clones, and 1 in clone 11393 from Ara+1), independent of IS186 transposition.

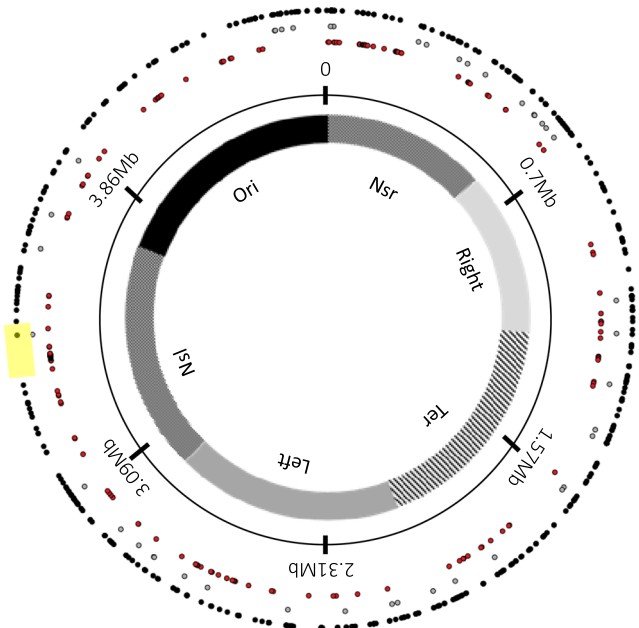

**Fig. 1 Distribution of all insertion sites of IS elements in the genome sequences of the ancestor and 264 evolved clones.** Insertion sites are mapped onto the chromosome of the LTEE ancestor, REL606. The inner circle indicates the six chromosomal macrodomains[41]: Ori origin; Nsr non-structured right domain; Right right domain; Ter terminus domain; Left left domain; Nsl non-structured left domain. The next circle shows the chromosomal coordinates. Gray and black dots show the location of IS elements in the ancestor and evolved clones, respectively. Red dots show the location of essential genes[20]. The 152-kbp "IS-empty" chromosomal region is highlighted in yellow. Source data are provided as a Source Data file.

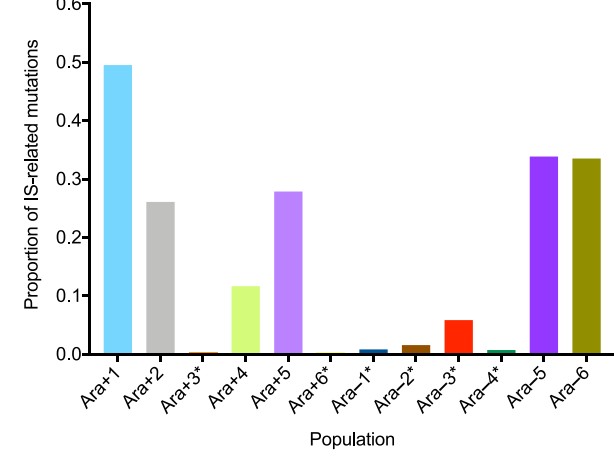

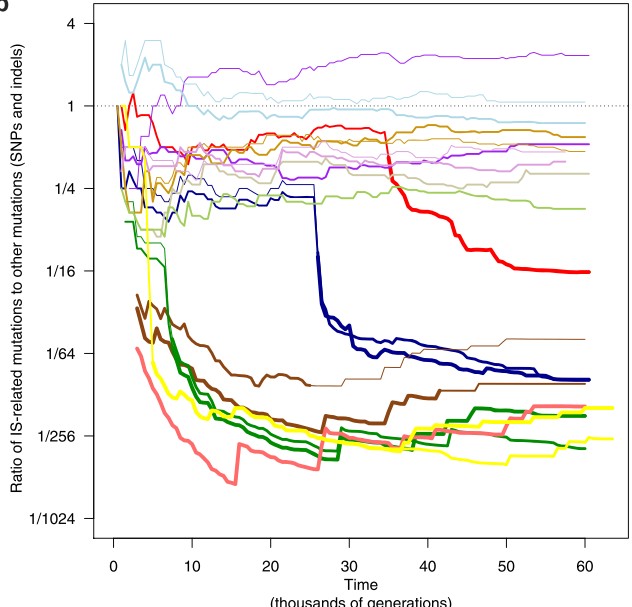

**Fig. 2 Effect of mutator phenotype on IS-related mutations. a** Proportion of IS-related mutations relative to total mutations after 50,000 generations. The numbers of mutations were inferred from genome sequences of evolved clones[18]. The ratios of IS-related mutations to total mutations were calculated for two clones from each population, and the data shown are the averages. Asterisks indicate six populations that evolved point-mutation hypermutability. **b** Ratio of IS-related mutations to other mutations (SNPs and indels) in all populations shown on a logarithmic scale. These data are based on metagenome sequences obtained from whole-population samples over time; they include only IS insertions and SNPs that eventually reached fixation in the entire population, or in a subpopulation when a stable polymorphism was detected. The same colors are used in **a** and **b** (including Ara+6, yellow, which is barely visible in **a**). Major and minor (when present) lineages are shown by thin and very thin lines in the non-mutator populations, and by bold and lighter lines in the mutator populations. Source data are provided as a Source Data file.

mutations that accumulated in each of the 12 populations during the LTEE.

We mapped the new insertion sites of all IS elements onto the ancestral chromosome (Fig. 1). All regions that were nearly devoid of IS elements in the ancestor were colonized by IS elements in one or more populations, including the replication origin and termination regions, except for a 152-kbp region from 3.315 to 3.467 Mbp. This region comprises ~3.3% of the ancestral chromosome; it contains a single ancestral copy of IS1 and 160 genes, including 30 that encode ribosomal proteins or RNA and 35 essential genes[20]. Thus, the high density of critical and even essential genes may have resulted in purifying selection against new IS element insertions in this region.

**Contribution of IS elements to total genomic mutations.** We calculated the proportion of all accumulated mutations that were produced by IS elements from both the genomic data of the individual clones sampled at 50,000 generations[18] (Fig. 2a) and the metagenomic data of the 12 populations through 60,000 generations[17] (Fig. 2b). We define IS-related mutations to include both IS insertions and IS-mediated recombination events. IS-related mutations account for <6% of all mutations in populations Ara+3, Ara+6, Ara−1, Ara−2, Ara−3, and Ara−4, with an average of ~1.5% (Fig. 2a and Supplementary Fig. 1). These six populations evolved hypermutable phenotypes owing to mutations affecting DNA metabolism and repair processes, which increased their point-mutation rates by roughly two orders of magnitude relative to the ancestor[2,3]. By contributing more mutations that increased the overall count, the evolution of point-mutation hypermutability greatly reduced the proportional contribution of

other types of mutations, including IS-related mutations. In the six populations that retained the low ancestral point-mutation rate, the proportion of IS-related mutations ranged from ~11% in Ara+4 to ~50% in Ara+1 after 50,000 generations, with an average of ~30%. The high proportion in Ara+1 resulted from the exceptional activity of IS150 in that population. These observations were confirmed by the metagenomic data (Fig. 2b),

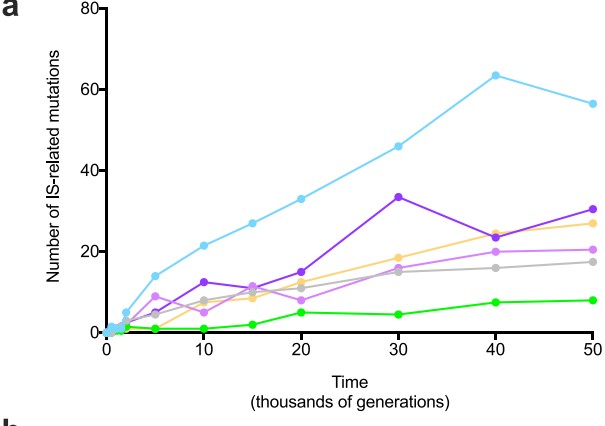

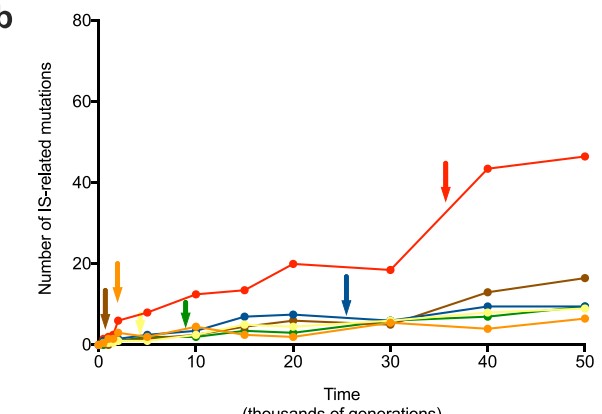

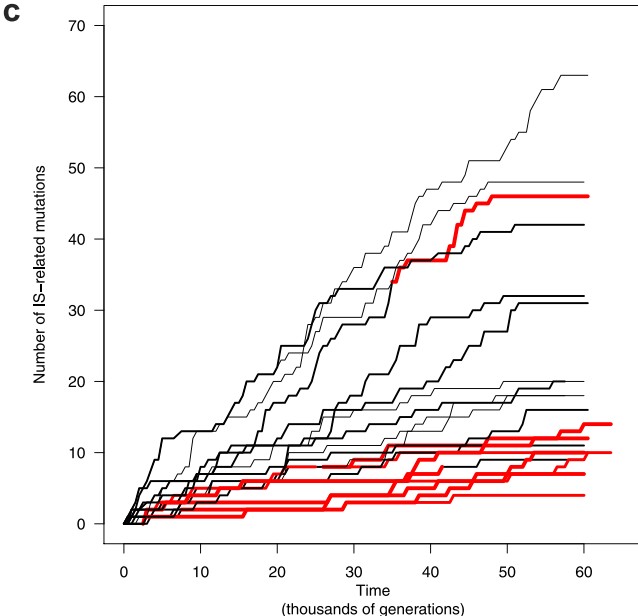

**Fig. 3 Dynamics of IS-related mutations in the LTEE populations. a** Number of IS-related mutations, compared to the ancestor, in the genomes of evolved clones sampled over time in the non-mutator populations[18]. **b** Number of IS-related changes in the genomes of evolved clones sampled over time in the mutator populations[18]. Arrows indicate the times of transition to the mutator phenotype. The color code for the populations is the same as in Fig. 2. **c** Number of IS-related mutations detected in the metagenomes of entire populations[17]. Trajectories include only those mutations that reached fixation in the entire population, or in a subpopulation when stable polymorphisms were detected. The number of IS-related mutations in major and minor subpopulations are shown as normal and thin lines, respectively, and mutator populations are shown in red after hypermutability evolved. Source data are provided as a Source Data file.

and Supplementary Fig. 1), with a tendency for fewer such changes in the mutators (Fig. 3c). This association is relatively weak, however, which may result from the accumulation of IS-related mutations while the point-mutation rate was still low in the mutator populations. In that case, the association would be confounded by mixing IS-related mutations that occurred under high- and low-mutation rates. To circumvent this limitation, we analyzed time series of the metagenomes of the entire populations.

**Effect of mutator state on the number of IS-related mutations fixed in the genome.** For each population we used the available metagenomic data[17] to infer whether mutations had occurred while it was in a mutator or non-mutator state (see "Methods"). We computed the number of generations of evolution under a high- or low-mutation rate and, over each period, the number of IS-related mutations that would eventually invade the population and reach fixation. The metagenomic data revealed the coexistence of multiple long-lived lineages in some populations. We performed our analysis by focusing either on the major lineage only or on both major and minor lineages when present. This analysis shows that fewer IS-related mutations fixed when the point-mutation rate was high (Fig. 4 and Supplementary Fig. 2). Taking into account all mutations found in the major lineages, there were 48 fixed IS-related mutations in 215,500 mutator generations, compared with 185 such mutations in 384,500 non-mutator generations. This difference is highly significant (two-proportion $z$-test, $p < 10^{-5}$), and it yields a ratio for the fixation rate of IS-related mutations per generation in mutator versus non-mutator lines of 0.46. If we include both major and minor lineages (69 fixed IS-related mutations in 351,500 mutator generations and 355 such mutations in 658,500 non-mutator generations, $p < 10^{-15}$), then the corresponding ratio is 0.36 (Fig. 4). These differences are even more pronounced, with ratios of 0.36 ($p < 10^{-9}$) and 0.27 ($p < 10^{-15}$), respectively, if we exclude the IS-related mutations that emerged around the same time that mutators evolved (Supplementary Fig. 2), because the mutations are difficult to attribute to one mutation-rate state or the other. Overall, these results show that the fixation of IS-related mutations occurred at roughly half the rate when the point-mutation rate was high as when it was low.

**Contribution of IS elements to fitness during 50,000 generations.** Given the large variation in the number of IS-related mutations in different populations, we asked whether it was correlated with the fitness improvements observed in these populations. We first used estimated fitness values at 50,000 generations based on the fit of power-law trajectories to data from

which show that the proportion of IS-related mutations began to decline when each specific population became hypermutable. Populations Ara+3, Ara+6, Ara−2, and Ara−4 all evolved hypermutability during the first 10,000 generations of the LTEE[3], whereas populations Ara−1[2] and Ara−3[21] did so after 20,000 and 30,000 generations, respectively.

However, this difference in the proportion of IS-related mutations between mutator and non-mutator populations was not caused solely by changes in the point-mutation rate. We also saw substantial variation in mutator and non-mutator populations in the absolute number of IS-related mutations (Fig. 3a, b

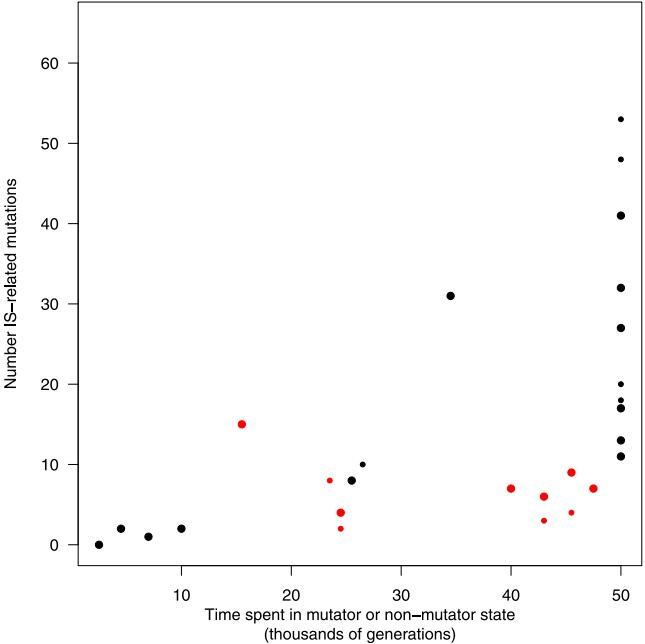

**Fig. 4 Number of fixed IS-related mutations as a function of a lineage's time spent in mutator or non-mutator state.** Large and small dots show mutations fixed in major and minor lineages, respectively. (Both include mutations fixed in the total population, so they are not fully independent). Red and black dots indicate mutator and non-mutator populations, respectively; the *y*-axis values show the number of IS-related mutations that were fixed over the time spent in either the mutator or non-mutator state. Source data are provided as a Source Data file.

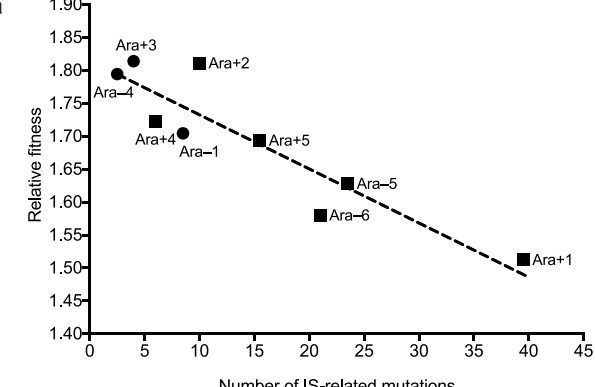

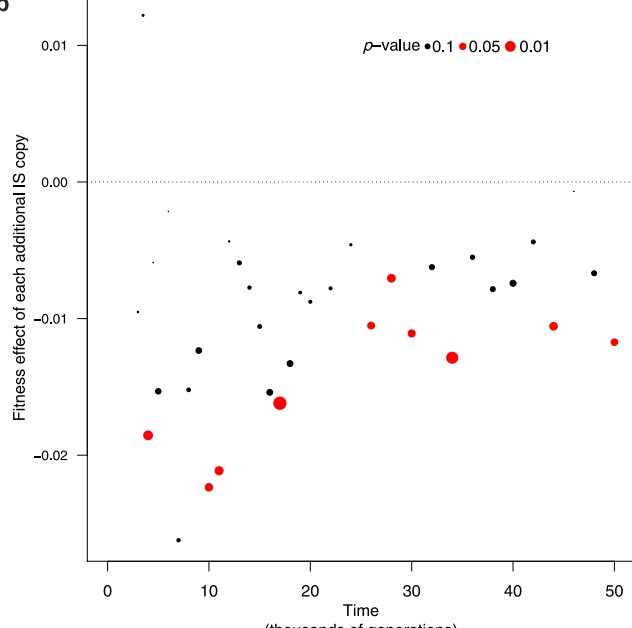

**Fig. 5 Relationship between fitness and IS-related mutations. a** Mean fitness of evolved populations at 50,000 generations, relative to the common ancestor, as a function of the number of IS-related mutations. The number of IS-related mutations uses the average value of the two evolved clones sequenced at that time point. Fitness values are inferred from a power-law model according to Wiser et al.[13]. Three populations (Ara−2, Ara−3, and Ara+6) are excluded owing to technical problems associated with measuring their fitness values[13]. Circles and squares indicate mutator and non-mutator populations, respectively. The line shows the least-squares linear regression $y = 1.813 − 0.0081x$ ($n = 9$, $r = −0.9115$, two-tailed $p = 0.0006$). **b** Multiple regression analysis to estimate the marginal fitness effect (the cost, when negative) associated with having an extra IS copy, taking into account the lineage's mutator or non-mutator status over time. The size of the points is larger for lower *p* values; red points indicate a significant contribution of IS copy number to fitness at the 5% level. Source data are provided as a Source Data file.

competition assays[13]. As in that previous work, we excluded three mutator populations (Ara−2, Ara−3, and Ara+6) for which technical issues precluded reliable assays in later generations. We detected a clear negative correlation between the fitness improvement in a population and the number of IS-related mutations (Fig. 5a, $r = −0.9115$, $n = 9$, two-tailed $p = 0.0006$). This trend remained significant even when considering only the six non-mutator populations ($r = −0.8844$, $n = 6$, $p = 0.0193$). This result suggests that higher IS activity was detrimental to adaptation, at least over the long run, indicative of an evolutionary constraint.

We then used the time series of metagenomic data[17] to quantify more precisely the impact of IS-related mutations on fitness. First, using all data, we performed a multiple regression to quantify a population's fitness as a function of the sampled generation, its mutator status at that time, and the number of IS-related mutations in the population. This analysis indicates that the number of IS-related mutations in a genome significantly reduced fitness (Supplementary Table 1, maximum $p < 10^{−8}$). This outcome was seen under several alternative models: whether generation was used as (1) a factor (treating each sampled generation independently); (2) a linear variable (fitness ~ a $N_{IS}$ + b G, where $N_{IS}$ is the number of IS elements and G the generation); or (3) transformed with a power law to take into account the non-linearity of the relation between fitness and time (fitness ~ a $N_{IS}$ + (b G + 1)$^c$). The same outcome was also seen whether or not we accounted for the population's mutator status, and whether or not we included the number of non-IS mutations (SNPs and indels not involving IS elements) in the model. Conversely, the number of SNPs and indels showed only a weak association with fitness when controlling for evolutionary time. In this analysis, which included only non-mutator populations, we found a marginally positive association between fitness and the

combined number of SNPs and indels when evolutionary time was included as a linear variable (slope = 0.0034, $p = 0.0293$).

Depending on the specific analysis, the estimated effect on fitness per IS copy ranged from −0.6% to −0.8% with standard errors of less than 0.1%. The intensity of the cost per IS copy is surprising: it implies that a genome that has 30 more copies than others bears a fitness cost of about 20%. To examine whether this fitness cost was consistent over time, we plotted the slope of the regression (i.e., the cost per additional IS copy) as a function of

time (Fig. 5b and Supplementary Fig. 3). Although the variance was higher in the early generations (owing to the small number of IS mutations, and perhaps also reflecting more mutations with large beneficial fitness effects), the average cost of each additional IS copy was reasonably constant over time.

To look more deeply into the fitness effects of IS-related mutations over time, we modeled fitness as a function of the numbers of IS-related mutations and other mutations (SNPs and indels), without accounting for the sampling generations (Fitness $\sim a\, N_{IS} + b\, N_{SNPs+indels}$). When the analysis is restricted to the first 4000 generations, fitness is positively correlated with the numbers of both IS-related and other mutations (slope = 0.014, $p$ value = 0.014 for IS, and slope = 0.011, $p$ value = 0.0001 for SNPs and indels). However, if we look across later samples (from 5000 to 50,000 generations), fitness is negatively correlated with the number of IS-related mutations (slope = −0.0043, $p < 10^{-4}$), while the correlation of fitness with the number of SNPs and indels remains significantly positive (slope = 0.0088, $p < 10^{-20}$ for SNP/indels). Finally, using a 10,000-generation sliding window, the number of IS-related mutations has a significant negative impact from time interval [14,000–23,000] to the final interval [41,000–50,000], whereas the number of SNPs and indels had a significant positive correlation with fitness up to interval [4000–13,000] and generally positive but non-significant effects over later 10,000-generation intervals. This analysis shows that early in the LTEE, both IS-related mutations and other mutations (SNPs and indels) that reached high enough frequencies to be observed contributed positively to adaptation. Relatively soon, however, IS-related mutations began to exert negative fitness effects, on average, while the average effects of SNPs and indels remained positive but faded in magnitude and statistical significance, owing to both smaller effect sizes of remaining beneficial "driver" mutations[13] and a higher fraction of non-adaptive "passenger" mutations[18]. Therefore, IS-mediated mutations first tended to promote and then, later to constrain, evolvability in the LTEE populations.

**Mutator status, IS elements, and evolvability**. As shown above, IS-related mutations often impose a fitness cost. However, previous studies have shown that a high point-mutation rate can increase the rate at which evolving populations adapt and increase their fitness[2,6,22]. We performed a multivariate analysis of fitness as a function of time and hypermutator status over the entire dataset. This analysis indicates that hypermutability provided a net fitness benefit of about 8–10% ($p < 10^{-10}$, Supplementary Table 1), in agreement with theory and previous analysis[13]. However, including the number of IS-related mutations in the analysis reduces the contribution of hypermutability by about half or even more (depending on how we account for time, Supplementary Table 1). This effect occurs because hypermutability was associated with a reduced accumulation of IS-related mutations, which in turn imposed a fitness burden. Thus, a plausible hypothesis is that a substantial proportion of the net benefit of evolving a high point-mutation rate derives from a reduction in IS-related mutations. Hence, the long-term benefits of point-mutation hypermutability may involve two factors: (1) increasing the genetic diversity that fosters adaptation; and (2) producing beneficial mutations through a mechanism that does not impose as high a cost as IS-related mutations, which are more likely to be selected in lineages with low point-mutation rates.

**Dynamics of IS-related mutations in Ara+1**. The Ara+1 population had both the most IS-related mutations (Table 1) and the lowest fitness gain of the 12 LTEE populations[13] after 50,000 generations. In particular, this population experienced an unusually high activity of IS*150* (Table 1), with ~89% of the IS-related mutations involving this element. The average number of IS*150*-related mutations in its 50,000-generation clones was 35, whereas the average for the other populations was only ~7.2. The unusually high activity of IS*150* in Ara+1 might be responsible for the atypically low fitness trajectory of this population. Therefore, we examined more closely the dynamics and activity of IS elements in this population.

From the genome sequences of 22 evolved clones sampled over time from population Ara+1, we computed the number of IS-related mutations (Table 1, Supplementary Data 1) and their proportion relative to total mutations (Supplementary Fig. 1). The number of IS-related mutations increased over time, but at a decelerating rate, such that almost 30% of them arose during the first 5000 generations (Fig. 3a, light blue line). As noted, most IS-related mutations involved IS*150* elements. The proportion of all mutations in population Ara+1 that were IS*150*-related exceeded 65% at 5000 generations, before stabilizing at roughly half in later generations (Supplementary Fig. 1). Most IS*150*-related mutations were transposition events, which led to an eightfold increase in the copy number of this element in this population (Supplementary Data 1).

Four hypotheses might explain this striking increase in the number of IS*150* copies. First, the sequence of one or more copies of IS*150* in population Ara+1 may have mutated to generate a hyperactive variant. Second, other mutations that occurred during the evolution of that population may have altered the regulation of IS*150*'s activity and thereby increased its transposition rate. Third, other mutations might have increased the cell's robustness to IS mutations, allowing more to accumulate by reducing their typically adverse fitness effects. Fourth, early IS*150* transposition events may have improved cell fitness, allowing the resulting clones to increase in abundance, thereby leading to a higher average copy number, higher levels of transposase, and increased rates of transposition—in effect, producing a positive feedback. These hypotheses are not mutually exclusive.

**Sequence of IS*150* elements in Ara+1**. We scrutinized the genome sequences of the clones sampled from population Ara+1, looking for any mutations that altered the sequence of any IS*150* element. We found only one mutation, an 11-bp deletion in one copy of *insJ* encoding the IS*150* transposase in the clone REL10451 from 30,000 generations. Given its timing, this mutation is clearly unrelated to the burst in IS*150* activity in this population. We carefully examined the possibility that even one copy of IS*150* might differ from the ancestral sequence by a single point mutation. Given that each of the two sequenced 50,000-generation clones harbored 40 copies of IS*150*, we looked for short-read variant sequences that had a frequency of at least 1/80 in either clone (i.e., any variant read with even half as many examples as one would expect if a single variant IS*150* copy were present in either clone). However, we found no variant reads that fulfilled this criterion. By contrast, we found 46 point mutations in other IS elements (IS*1*, IS*2*, IS*3*, IS*4*, IS*30*, and IS*186*) in other populations, which shows that such mutations are detectable. Thus, mutations in IS*150* itself cannot explain the hyperactivity of these elements observed in population Ara+1.

**Transposition frequencies of IS*150* in Ara+1**. We measured the transposition rate of IS*150* by introducing the reporter plasmid pFDX2339[23] into the ancestral strain and six evolved clones from population Ara+1 (Supplementary Data 1), including one each from generations 2000, 20,000, and 30,000 (REL strains 1158C, 9282B, and 10450, respectively), and three from generation 40,000 (11008, 11009, and 11010). This plasmid carries a

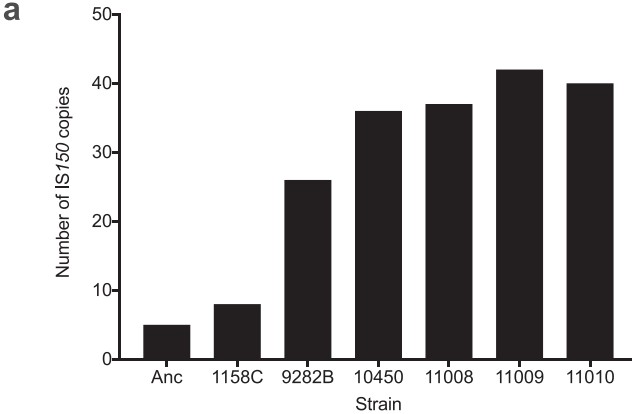

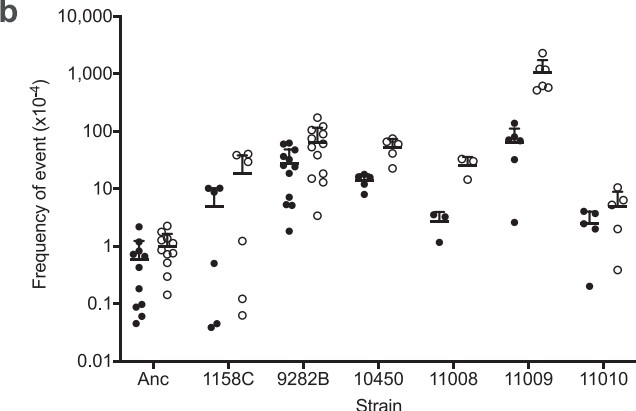

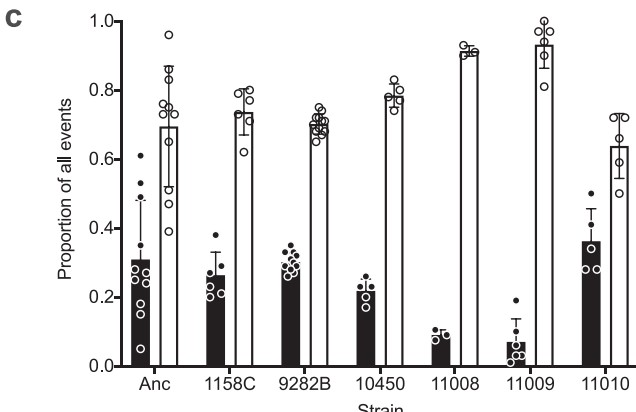

**Fig. 6 IS150-mediated recombination and transposition events in the ancestor and evolved clones from population Ara+1. a** Number of IS150 copies in genomes of the ancestor (Anc) and six evolved clones: 1158C (2000 generations), 9282B (20,000 generations), 10450 (30,000 generations), and 11008, 11009 and 11010 (40,000 generations). **b** Frequencies of IS150-mediated transposition (black) and recombination (white) events, measured using the reporter plasmid pFDX2339. **c** Proportion of transposition (black) and recombination (white) events, expressed relative to the total number of events (including both transposition and recombination). Error bars show the standard deviation based on three technical replicates for each of three biological replicates, except for the ancestor which had three technical replicates for each of five biological replicates and 1158C and 11009, which had two biological replicates. Source data are provided as a Source Data file.

modified IS150 element that enabled us to estimate the frequencies of both IS150 transposition from the plasmid into the chromosome and recombination between the plasmid-borne IS150 and any chromosomal copy of that element. The latter category served as a proxy to account for the increase in IS150 copy number on the evolved chromosomes (Fig. 6a). As expected, both transposition and recombination frequencies were higher in the evolved clones than in the ancestor (Fig. 6b). The rates of both processes increased much faster than the number of chromosomal IS150 copies through the 20,000-generation sample (Supplementary Fig. 4). This outcome was particularly surprising for the frequency of recombination events, which we expected to scale with the number of target copies on the chromosome.

We do not know why the recombination frequency increased faster than the number of target copies on the chromosome. One could imagine, for example, that this trend might reflect changes in DNA topology. In fact, most LTEE lines evolved altered supercoiling via one or more mutations affecting the *topA*, *fis*, or *dusB* (*yhdG*) genes[24]. The Ara+1 population had a nonsynonymous mutation in *topA* that reached high frequency before generation 2000, but which did not achieve fixation, and a later mutation in the promoter region upstream of the cotranscribed *dusB* and *fis* genes that fixed by generation 15,000. However, it is unclear whether these particular mutations somehow contributed to the atypical burst in IS150 activity seen in this population or to changes in its transposition rates. In any case, the relative rates of transposition and recombination events were similar in the evolved clones sampled through generation 30,000 to what they were in the ancestor (Fig. 6c). Given the fairly stable ratio of transposition and recombination events, there is no compelling evidence for any change in the regulation of IS150 transposition per se during the first 30,000 generations that could explain the increase in IS150 copy number.

However, the ratio of IS150 transposition to recombination declined markedly in two (11008 and 11009) of the three 40,000-generation clones (Fig. 6c), despite the continued increase in the number of chromosomal IS150 copies through 40,000 generations (Fig. 6a). For evolved clone 11008, this change reflected a drop in transposition events, both in absolute numbers (Fig. 6b) and, especially, on a per-copy basis (Supplementary Fig. 4). For evolved clone 11009, both transposition and recombination frequencies actually increased, but the latter more so (Fig. 6b), leading to a lower ratio of transposition to recombination events (Fig. 6c). The 40,000-generation clone 11010 showed yet a third pattern, with reduced frequencies of both transposition and especially recombination events (Fig. 6b), which resulted in per-copy transposition and recombination rates that were most similar to the ancestral strain (Supplementary Fig. 4). Thus, following the increases in both IS150 copy number and transposition rate through 30,000 generations, the transposition rate later declined relative to recombination events in two of the three 40,000-generation clones, while both transposition and recombination rates declined in the third clone. These changes, coupled with the absence of mutations affecting the IS150 sequence per se, imply that one or more chromosomal mutations arose after 30,000 generations that reduced the activity of the IS150 elements. This reduced activity may have been coincidental (e.g., a correlated response to selection on some other function), or it may have conferred a direct benefit by reducing the rate of deleterious IS150-mediated mutations. Consistent with the latter possibility, it is intriguing that the early rise and later reduction in IS150 activity mirrors what has been observed for point-mutation rates in other LTEE populations[2,17,18].

**Fitness changes associated with early IS150 dynamics in Ara +1**. After only 2000 generations, an evolved clone, 1158C, sampled from population Ara+1 had already accumulated three new IS150 insertions (Supplementary Data 1) and two IS150-mediated deletions[18]. Two of the insertions and one of the deletions were also found in the other sequenced clone, 1158A, from that generation, which also had another IS150 insertion. These IS150-mediated mutations accounted for 5/9 and 4/9 of the total mutations found in these two clones. Moreover, all three shared IS150 changes would eventually reach fixation in that population[18]. The shared deletion involved an IS150 insertion, followed by homologous recombination with an existing IS150 element, that removed several *rbs* genes involved in ribose consumption; deletions of those genes confer a fitness advantage of ~1–2% in the glucose-limited conditions of the LTEE[15]. A new insertion IS150::*ybeB* in clone 1158C (but not present in 1158A) also went on to fix. It was shown to reduce transcription of the *pbpA* gene, which encodes the PBP2 transpeptidase involved in cell-wall biosynthesis, and that insertion conferred a fitness advantage of ~5–6% relative to an isogenic construct without the insertion[25]. Another shared IS150 insertion occurred in the *nadR* gene, and it presumably knocked out that gene's function. Inactivation of *nadR* has been shown to confer a fitness benefit of ~8% relative to the ancestral allele[26]. These three IS150-mediated mutations thus account for more than half of the fitness gain of clone 1158C, which was previously estimated to be ~24%[25]. At least one point mutation in this clone, that in *pykF*, was also beneficial[26,27]. With respect to the dynamics of IS elements, at least two early IS150 insertions in population Ara+1 were highly beneficial, and the resulting additional copies of the transposase-encoding gene could potentially drive the subsequent increase in transposition activity observed in this lineage.

## Discussion

Several decades after the pioneering work of Barbara McClintock on transposable elements in maize[28], Chao and McBroom showed a mutator-like benefit for IS elements in bacteria[29], while Charlesworth et al.[30] and Levin[31] discussed how mobile genetic elements could be either genetic mutualists or parasites. Today, the evolutionary forces that maintain mobile genetic elements remain a matter of debate. Given that IS elements can increase their copy number within a host genome, it is unclear what prevents them from increasing so much that they eventually drive an infected lineage to extinction. Previous work has suggested that bursts in transposition activity are deleterious and hence disfavored by natural selection[32], which could provide a counterbalancing force in evolving populations.

In this study, we followed the dynamics of IS elements in 12 populations of *E. coli* during 60,000 generations of experimental evolution. We found substantial variability among populations in the contribution of IS elements to their accumulated genomic changes, on both absolute and relative bases. The average absolute number of IS-related mutations in sequenced 50,000-generation clones varied more than 15-fold, ranging from just 2.5 in population Ara−4 to 39.5 in Ara+1. The proportion of genetic mutations that were IS-related varied over 300-fold, from ~0.1% in Ara+6 to 35% in Ara+1, with the relative contribution of IS-related mutations much lower in six populations that evolved hypermutability than in six others that retained the ancestral point-mutation rate. We found no direct correspondence between the ancestral copy number of a given IS element and its activity during evolution. For example, IS1 had 28 copies in the ancestor, and that number remained between 25 and 33 in the sequenced 50,000-generation clones. Similarly, IS3, which had 5 ancestral copies, varied in number between 3 and 8 in those same clones.

By contrast, IS150 also began with 5 copies, but its copy number ranged all the way from 5 to 40 in the 50,000-generation clones.

In the LTEE, IS150 was the most active element, particularly in the Ara+1 population. The number of IS150 elements increased eightfold in this population. This element was most active early in this population's history, with an average of 11.5 IS150-related mutations in the sequenced clones between generations 1500 and 5000, corresponding to ~3.3 per 1000 generations. There were another 5.5 mutations, on average, between generations 5000 and 10,000 (1.1 per 1000 generations), with an additional 18 from 10,000 to 50,000 generations (0.45 per 1000 generations). A few other populations showed bursts of IS150-related mutations that began later and were of shorter duration, including Ara−5 between 15,000 and 30,000 generations (0.9 per 1000 generations) and Ara−3 between 30,000 and 40,000 generations (1.6 per 1000 generations).

We closely examined the sequences of IS150 elements in all 22 sequenced clones from the Ara+1 population to see whether mutations in the element itself contributed to its altered activity. Other than a small deletion within one IS copy in one clone, we found no mutations. Instead, it appears that other processes must explain the initially increased activity followed by the subsequent decline. Three of the early beneficial mutations that went on to fixation in this population involved IS150 elements, including two insertions that generated additional copies. Those additional copies would increase the concentration of transposase, all else equal, and thereby increase the potential for further IS150 activity, as indeed occurred. It is also possible that evolved changes in global supercoiling or binding of nucleoid-associated proteins in Ara+1 modulated IS activity in this population, or that the local chromosomal context of one of the early insertions rendered that new copy hyperactive.

In addition to producing occasional beneficial mutations, however, a higher rate of IS element activity would also generate more harmful mutations. We observed a later decline in IS150 activity in the Ara+1 population that resulted from genetic changes elsewhere in the bacterial genome, which would have reduced this cost. It is unclear which of the many tens of mutations that accumulated in that population were responsible for the reduced activity of IS150 in later generations, but the resulting change suggests a sort of domestication of this element by the bacteria[9]. It is also possible that some of these mutations may have reduced the cells' robustness to IS activity, thereby limiting their accumulation. In addition to trans-acting mutations, the declining rate of IS150 transposition might reflect inhibition caused by increased production of a putative IS transposase regulator encoded by IS150 itself[23], as has been reported for IS3[33]. In any case, the changing dynamics of IS150 in Ara+1 are reminiscent of the dynamics of point-mutation rates in several other LTEE populations, which involved large increases in mutation rate followed by reversions in some lineages and compensatory changes in others[2,17,18]. These later reductions in point-mutation rates thus limited further increases in the genetic load caused by deleterious mutations[2].

The Ara+1 population, with its hyperactive IS150 elements, has also had the lowest fitness trajectory of any of the LTEE populations[13]. The genetic load is probably greater for IS-mediated hypermutability than for point-mutation hypermutability because transpositions are more likely to disrupt gene functions than point mutations. In addition, some IS-related mutations occur at exceptionally high rates[15]. Population Ara+1 not only had the lowest fitness of any population at 40,000 generations, it also exhibited the smallest fitness gain from 40,000 to 60,000 generations[34]. This pattern contradicts the tendency for less-fit genotypes to have greater scope for improvement and faster adaptation[13,34]. Thus, IS-element hyperactivity not only

increases the load of deleterious mutations, it may also constrain later evolvability. To understand this constraint, consider the following hypothetical case. Imagine a gene product for which a complete loss of activity would provide a mutant with a 5% fitness advantage in some new environment, while a 50% loss of activity would confer a 10% benefit. A complete loss of activity could be easily achieved by an IS insertion. But after an insertion, it would be difficult for a subsequent mutation—either in that same gene or one encoding a regulator of its expression—to restore partial activity. By contrast, a point mutation might achieve the new optimum in a single step; even if it did not, it would be easier for a secondary mutation to fine-tune the activity. Also, an increased number of IS elements in a genome may have an indirect fitness cost because IS copies are prone to recombine with one another, thereby generating deletions, inversions, and other rearrangements[35]. As more copies of a given IS element accumulate in a genome, the number of IS pairs that can recombine to generate deleterious mutations increases in a combinatorial fashion that compounds this cost. Taking all of the findings from population Ara+1 together, IS-element hyperactivity promotes short-term adaptation, but at the cost of more deleterious mutations and constraints on subsequent evolvability.

Our analysis of IS-related mutations also suggests a new, indirect benefit associated with the evolution of a high point-mutation rate. The rate of IS-related mutations can be much higher than that of typical point mutations[15]. Hence, in a population with a low point-mutation rate, such as Ara+1, IS-related beneficial mutations will occur earlier and may be fixed before the phenotypically equivalent point mutations. By contrast, in hypermutator populations, these IS-related beneficial mutations would likely compete with phenotypically equivalent or better point mutations that also appear at high frequency. Thus, fewer IS-related mutations would be fixed in populations with point-mutation hypermutability, as our analyses showed. Moreover, IS-related mutations may be autocatalytic, with most secondary events deleterious; and even when they are beneficial, IS-related mutations may often constrain subsequent opportunities for refinement. Hence, point-mutation hypermutability may confer a long-term advantage not only through a higher rate of beneficial point mutations, but also by precluding IS-related beneficial mutations from reaching fixation, with their associated secondary costs.

Many decades after their discovery, the evolutionary forces driving the maintenance and expansion of IS elements in bacterial genomes remain the subject of debate[30–32]. These elements have played major roles in shaping genomes. On the one hand, they have facilitated the extreme genome reduction that has occurred in endosymbionts over millions of years[31,36]. In these organisms, genetic drift caused by extreme population bottlenecks is thought to have reduced the power of selection to remove deleterious mutations, allowing IS elements to proliferate and thereby accelerate the loss of chromosomal genes via IS-mediated deletions. On the other hand, IS elements are often passengers during, and sometimes also enablers of, the horizontal transfer events that allow bacteria to acquire new functions by expanding their genomes.

In this study, we examined the dynamics of IS elements in an experiment spanning 60,000 bacterial generations. We found a multifaceted tension involving their potential to generate beneficial mutations while also producing deleterious mutations and constraining compensatory adaptations. A particular IS family generated highly beneficial mutations early in one evolving population, leading to an increase in its copy number and transposition rate. This family eventually evolved a lower transposition rate, but not before the population had accumulated deleterious mutations and constraints on its subsequent evolvability. Neither the early increase in transposition rate nor its later reduction was caused by mutations in the IS element itself. Instead, the increased rate appears to have been driven by the positive feedback between copy number and transposition, whereas the subsequent decreased rate resulted from chromosomal mutations that repressed transposition through undetermined mechanisms. It is also possible that changes in the robustness of cells to IS activity contributed to the different rates of accumulation of these elements in that population over time. Of course, another force affecting the dynamics of IS elements in nature—one that does not act in the LTEE—is horizontal transfer via conjugative and other transmissible elements. It would be interesting and useful, therefore, to design and perform long-term experiments that incorporate that process as well. In the meantime, our analysis of the LTEE shows how mutation and natural selection operating in large asexual populations can lead to both subtle and dramatic changes in the activity of IS elements that, in turn, affect the evolvability and robustness of bacterial genomes.

## Methods

**Bacterial strains and culture conditions**. The twelve populations of the LTEE, called Ara+1 to Ara+6 and Ara−1 to Ara−6, were founded from a common ancestral strain, *E. coli* REL606[37]. Since 1988, they have been serially propagated in a minimal medium (DM25) containing 25 μg/ml glucose[38]. Samples were taken from each population at 500-generation intervals and stored at −80 °C. In this study, we analyzed metagenomic data from population samples through 60,000 generations[17], together with previously sequenced genomes of the ancestral strain and two evolved clones sampled from each population at each of 11 time points[18]: 500, 1000, 1500, 2000, 5000, 10,000, 15,000, 20,000, 30,000, 40,000, and 50,000 generations (Supplementary Data 1). In experiments described below, some of these strains were grown in LB broth with orbital shaking at 200 rpm or on LB agar (12 g/l) plates at the stated temperature. When indicated, the bacteria were grown in LB media supplemented with kanamycin (Kan, 30 μg/ml), chloramphenicol (Cam, 40 μg/ml), or both.

**Measurement of IS150 transposition frequencies**. Cells from the ancestor (REL606) and six evolved clones from population Ara+1 (REL isolates 1158C, 9282B, 10450, 11008, 11009, and 11010) were electro-transformed with the pFDX2339 reporter plasmid[23] using the GenePulser II equipment (BioRad), following standard procedures. This plasmid has a thermo-sensitive replication origin (allowing maintenance at low but not high temperature), a kanamycin-resistance gene (allowing selection for plasmid-carrying cells), and a modified IS150 element bearing a chloramphenicol-resistance gene. When cells are grown at high temperature, there are several possible outcomes: (1) the plasmid and its modified IS150 may be lost; (2) the plasmid may integrate into the chromosome by recombination between its own IS150 copy and a chromosomal IS150 element; or (3) the IS150 from the plasmid may transpose into the chromosome, with the rest of the plasmid being lost. These outcomes can be distinguished by plating cells on different selective media: cells that are sensitive to both chloramphenicol and kanamycin indicate loss of the plasmid and its IS copy; cells resistant to both chloramphenicol and kanamycin correspond to recombination events; and cells resistant to only chloramphenicol imply a transposition into the chromosome.

Each plasmid-carrying strain was grown in 2 ml LB + Kan at 28 °C for 48 h, diluted and spread onto two sets of agar plates, LB + Kan + Cam and LB + Cam, which were incubated at 28 °C and 42 °C, respectively. We scored the total number of cells carrying pFDX2339 based on the number of colonies on the first set of plates, and the total number of cells that integrated into the chromosome—either the entire plasmid by recombination (R) or only the modified IS150 copy by transposition (T)—based on the number of colonies on the second set. To distinguish between the latter two events, colonies that grew at 42 °C on LB + Cam plates were streaked onto both LB + Kan + Cam and LB + Cam plates and incubated at 42 °C. Colonies that grew after streaking only on the LB + Cam plates gave T, while those that grew on both plates gave T + R, allowing us to distinguish cells that experienced IS150-mediated transposition and recombination as T/(R + T) and 1 − T/(R + T), respectively. Each strain's reported value is the mean of at least three technical replicates for each of three biological replicates.

**Location of IS elements on the chromosome**. Based on *breseq*-enabled analysis of the genome-sequence data[18], we mapped the locations of IS elements for each LTEE population on the ancestral REL606 chromosome using the Civi software (version 1.0)[39].

**Attribution of mutations to low or high mutation-rate status**. To assess whether mutations occurred in a lineage with a high point-mutation rate, we used the timing of each mutation's occurrence and fixation in the lineages found in each of

the 12 populations[17]. We first tested for the presence of coexisting lineages in a population and, if they were found, estimated their frequencies over time. Next, we used a hidden Markov chain to compute for each mutation at each sampled time point its most likely state among: ancestral, polymorphic, fixed or lost in the whole population; and if polymorphic, fixed or lost in the major or minor lineages when present. We also characterized the times of appearance and fixation, as well as the potential time of loss, of point-mutation mutator alleles in each population. A mutation was counted as having occurred in a non-mutator background either if it was detected before the mutator allele appeared or if it fixed after a mutator reversion had emerged. Conversely, a mutation was considered to have occurred in a mutator background if it was detected at the same time as or later than the mutator's appearance and it had fixed before reversion of the mutator was detected. However, it is not possible to attribute every mutation to a mutator or non-mutator background with certainty, because some mutations occurred around the same time as the transitions of mutation rates. Therefore, we also performed the analysis excluding mutations that appeared within 1000 generations before or after a mutator allele appeared, and those that fixed within 1000 generations before or after a mutator reverted. Finally, to avoid any bias due to the fixation times of mutations, we analyzed the data only through 50,000 generations, such that an additional 10,000 generations remained for mutations to fix after the last point analyzed.

**Software**. Microsoft Excel (versions 2013 and 2016) was used for data analyses. Breseq[18] version 0.25d and Civi[39] version 1.0 were used for genome analyses. Figures 2a, 3a, 5a, b, and 6 were plotted using Prism version 8.4.3. Regression models using the function lm from R software (RStudio version 1.1.453 and R version 3.6.2 2019-12-12) were used for analyzing the relationship between IS elements, mutator status, and fitness.

**Reporting summary**. Further information on research design is available in the Nature Research Reporting Summary linked to this article.

## Data availability
The datasets produced by this study have been deposited[40] to the Dryad Digital Repository (doi:10.5061/dryad.m37pvmd0v). Source data are provided with this paper.

## Code availability
The code used to analyze the genomic and metagenomic datasets has been deposited as a Rmd file to the Dryad Digital Repository[40] (doi:10.5061/dryad.m37pvmd0v).

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

## Acknowledgements
This work was supported by the French Centre National de la Recherche Scientifique (CNRS, to D.S.), the University Grenoble Alpes (to D.S.), the French CNRS International Associated Laboratory (to D.S and R.E.L.), the European Commission 7th Framework Programme (EvoEvo Project ICT-610427, to D.S.), the French Agence Nationale pour la Recherche ANR GeWiEp (ANR-18-CE35-0005-01, to D.S. and O.T.), the French Fondation pour la Recherche Médicale (EQU201903007848, to O.T.), the US National Science Foundation (grant DEB-1951307, to R.E.L. and J.E.B.), and the US Department of

Agriculture (MICL02253 to R.E.L.). We thank Bodo Rak for giving us the pFDX2339 reporter plasmid.

**Author contributions**
J.C. and J.G. performed all experiments; O.T. performed the analyses of the metagenomics data; J.E.B. and O.T. provided genomic sequences; J.G., R.E.L., T.H., O.T., and D.S. analyzed data; J.G., R.E.L., O.T., and D.S. wrote the paper. All authors edited the paper.

**Competing interests**
The authors declare no competing interests.

**Additional information**

