## [Peer Review File · Nature Communications]

Reviewers' Comments:

Reviewer #1:

Remarks to the Author:

The authors present a novel set of analyses of the role of IS mutations during 50,000 generations of experimental evolution of 12 bacterial populations. They show that IS mutations are common, making up to 35% of all mutations in some populations, but that their contribution is two-fold lower in populations with increased point mutation rates (mutators). Further, by exploiting high-resolution genome sequence and fitness data for these populations, they estimate that these IS mutations have on average a small negative effect on fitness, while previous work showed that early IS mutations can also have substantial positive effects. Most remarkably, populations with many IS-related mutations reach lower fitness improvements, suggesting that IS mutations constrain long-term evolvability. The authors also use an elegant assay with a reporter plasmid to estimate rates of IS activity in genotypes from different time points from a population with exceptional IS activity, showing an initial acceleration of IS mutations relative to the increase in number of IS copies, followed by a decline due to mutations outside the IS elements. Together these analyses show that, while IS-mediated mutations may sometimes drive adaptation, they also constrain adaptation in the long term and hypermutability status may help to limit the negative impact of IS mutations by providing alternative, perhaps superior, adaptive variation.

I would like to see this study published in *Nat. Comm.*, but also have a few questions and suggestions, which are hopefully helpful for conveying their message.

1. In my opinion, the main advance of this work is the message that IS-mediated mutations, while sometimes adaptive themselves, put constraints on long-term evolvability. But this message was a bit hidden, certainly in the title (which I find little informative), but also in abstract and text. I would prefer a stronger emphasis on this main message, but I realize that this remains also a matter of taste.
2. In all analyses, except for population Ara+1, no distinction is made between IS insertions and IS-mediated recombination events causing structural variants. It would be instructive to separate these two classes of IS mutations, because one (IS insertions) requires IS activity, while the other does not necessarily require this (recombination may happen between existing copies). It would also be good to define early on what you mean with IS mutations, as it took me some time to realize that both types were included.
3. The regressions used to estimate the fitness effects of IS mutations combine the number of IS mutations with a factor representing time. I found this a bit awkward and expected one or more other factors to explicitly represent the number of other mutations (point mutations, indels). Were such regressions also tried, and if so, what estimates did they yield? Similar estimates for the fitness effects of point mutations and indels would be very helpful for testing whether the reduction of IS mutations in mutator populations was driven superior, or simply more, mutants with point mutations/indels. Also, would it be possible to incorporate a simple form of epistasis, such as diminishing returns epistasis, in these regressions?
4. Related to the previous comment, would you have the statistical resolution to ask whether early IS-mutations across all populations (not only Ara+1) might have positive fitness effects, and negative effects only dominate later in evolution? This might help to generalize the adaptive dynamic perspective sketched for Ara+1.
5. With respect to the reporter assays to measure the frequency of IS150 mutations in genotypes from population Ara+1, could it be that measured frequencies not only reflect transposition and recombination rates, but also differences in selection? For example, the decline in frequencies after 20,000 generations (Fig. 6b) might also be due to decreased robustness to IS mutations. Perhaps the authors can simply ignore this based on previous work with these lines, but otherwise it might be useful for the readers to discuss this possibility.
6. On p13, it is speculated that mutations DNA topology, including the *fis* operon and *recF*, may be involved in the early upregulation of IS-150 activity. Were these regions particularly affected by

mutations in Ara+1? Similarly, on p16 (bottom) it is mentioned that transposase concentration is involved in regulating IS activity, but it is unclear how transposase is regulated (or rather constitutively expressed).

7. In Fig. 4 it is not clear to what time interval these numbers of IS mutations relate (cumulative or only new IS mutations, e.g. 1,000-generation intervals?).

Reviewer #2:

Remarks to the Author:

Review Consuegra et al.

The work by Consuegra et al. is an absolutely outstanding manuscript that provides experimental data on IS evolution that have been theoretically discussed for decades. The statement in the authors' abstract "The long-term dynamics of IS elements and their effects on bacteria are poorly understood" is in my opinion very true. It was until now very unclear how insertion sequences persist in bacterial genomes and what their long term effects are on bacterial genome evolution.

As the authors have noted there has been a long standing debate on what evolutionary forces maintain insertion sequences or other transposons in bacterial genomes. Is it only horizontal transfer between genomes that allows transposons to persist inside the gene pool or do the rare beneficial mutations they cause (one could call an IS element a mutator gene) significantly extend their persistence. To address these questions the authors analyze the insertion sequence content of 12 independent evolution lines of the long term evolution experiment in *E. coli* B over the period of about 60,000 bacterial generations. Their findings are quite extraordinary. Almost every section of the paper adds significantly to our understanding of insertion sequence evolution and its impact on bacterial evolution.

First, the ancestral IS copy number does not predict IS activity (new insertions) in the experiment. This was completely unexpected to me, but may make sense if large IS families have already been silenced by the host genome (something that could be investigated in future studies). Moreover, different IS elements showed different level of activity (new insertions) across the 12 lines. The authors present evidence that this may be due to historical contingency. If an insertion sequence has caused a beneficial mutation early on, this IS element will be maintained and simultaneously the transposition rate of the IS family will increase due to an increase in IS copy number.

Furthermore, the authors offer several other – maybe even more significant – insights into IS element evolution:

1. IS mutations occurred at twice the rate in non-mutator strains compared to mutator strains.
2. The fitness cost of carrying a single IS element is about 0.6%. Carrying 30 elements reduces your fitness by about 20%. This means carrying IS elements is costly. A basic assumption in most theoretical models, which finally has been experimentally tested at a large scale.
3. In contrast to carrying IS elements being a hypermutator is beneficial.
4. The authors find that there seems to be a negative correlation between the accumulation of IS elements and the evolution of high mutation rates. IS accumulation prevents the evolution of high mutation rates or vice versa. I am quite interested in the causalities of this relationship. As the authors have mentioned, a high mutation rate seems to reduce the transposition rate, but at the same time a high transposition rate reduces the need for a high mutation rate. I think this is a very interesting observation and I can't wait to see this modelled in future studies.

There are a few minor points in the manuscript that the authors could address, discuss in the manuscript or simply reply to in the reviewer response. The manuscript is already full of very interesting data and discussion that it really should be up to the authors whether they want to add anything else or whether they prefer to leave the manuscript as is.

1. The authors suggest that once IS elements have caused a beneficial mutation the bacterium may be stuck on a suboptimal local adaptive peak. This means at least two mutations are needed to reach a different peak (i.e. first removing the IS elements, which reduces fitness, and then another mutation to reach a different peak). These local optima seem to be quite stable since the authors have not reported any IS element excisions or deletions, and I imagine even if an IS element excises, then the excision is in most cases not scar-less. Could the scar leaving mechanism be a way for an insertion sequence to increase its persistence in the bacterial genome (excision of the IS element is not achieved by the transposition process itself)? How common are IS element deletions (are there genomic deletions that involve IS elements)? Could you estimate the persistence time of IS elements in bacterial genomes based on the available data?

2. Is there evidence of horizontal transfer between strains in a single line? I could imagine that biased gene conversion mediated by the IS element could lead to persistence in the genome, when a subpopulation loses the element. Is it possible that this could be happening within an evolution line?

3. What would happen if two lines were mixed (one containing lots of IS elements and one without)? The line with lots of IS elements would probably quickly die out. However, it is possible that biased gene conversion could lead to some of the IS elements moving into the genomes lacking IS elements. Has anything like this been attempted? How would IS elements fare if the LTEE environment were structured (i.e. the cultures are mixed every 5000 generations or similar)? Would you see more or fewer strains with an increase in IS element activity?

3. Something I found rather confusing is the discussion of the transposition rate per element for IS150. The way I understood the problem, is like the author put it in the last paragraph: "the increased rate [per element transposition rate] appears to have been driven by the positive feedback between copy number and transposition". But earlier in the manuscript when the authors measured the transposition rate of IS150 in Ara+1 they argued the opposite point "However, the rates of both processes increased much faster than the number of chromosomal IS150 copies through the 20,000-generation sample (Supplementary Fig. 4), whereas we had expected that the frequency of recombination events, at least, would scale with the number of target copies on the chromosome.". From my vantage point I can interpret the last sentence in two different ways, both of which are confusing.

First, the authors believe that there should be a linear relationship between transposon copy number and transposition rate. I don't think a linear relationship is plausible (unless I misunderstand something). If we assume a simple reaction norm where a transposon gene produces a transposase enzyme at rate k_1 : $T \rightarrow T+E$ and a transposase enzyme copies a transposase gene at rate k_2 : $T+E \rightarrow 2T+E$ then we get the following differential equations that describe the change in gene number and enzyme number: $dE/dt = k_2 * T$ and $dT/dt = k_1 * E * T$ if we solve these equations we can calculate the concentration of E and T at different time points: $T(t) = T(0) * \exp(k_1 * t * E)$ and $E(t) = k_2 * T * t + E(0)$. Hence, we would see an (over)exponential relationship between transposition rate (the number of transposition events measured in the author's experiment) and encoded transposon genes without changing the per element transposition rate k_2 (to explain the data one could for example set k_1 to 1 and k_2 to 1/20 or similar).

Second, I could interpret "scale" to mean that the number of transposon genes scales in an exponential way with the transposition rate. But in this case there is no conflict with the data.

It is possible that I have misunderstand something significant here, but to avoid confusion, I think it would help if the authors could clarify how they believe the number of transposon genes should scale with the transposition rate (i.e. present a simple model that suggests this relationship) and then explain how the experiment does not agree with this model.

4. I do not like the data availability statement. I am sure the authors will provide the data/code on demand and that they will make it available after publication. However, I have come across too many papers where it has been incredibly difficult or impossible to obtain data/programs from the author after publication. It would be nice if the authors could make all data available publicly before publication of the manuscript. I also think this usually makes the paper more accessible and hence may lead to more citations.

5. Finally, a very minor request. There are some sentences in the introduction, for which citations are missing. It would be nice if the authors could provide those. See for example: lines 31, 41, 54-59 and 69.

6. line 197/198: "a factor (treating each sampled generation independently);", what does this mean?

Frederic Bertels

Point-by-point response to the reviewers' comments on the NCOMMS-20-33123 manuscript titled "IS-mediated mutations both promote and constrain evolvability during a long-term experiment with bacteria."

We thank the editor and reviewers for their helpful comments and constructive feedback. We have addressed all their comments, and we think that our revised text is much improved as a result.

Briefly, we edited our manuscript to convey our results and interpretations in light of the reviewers' comments, as follows:

- We emphasized the main message of our manuscript at several places, including by changing the title to make clear that IS-mediated mutations, while sometimes adaptive, can also constrain long-term evolvability.
- We defined clearly what we mean by IS-related mutations.
- We performed the suggested analysis in which we also include the number of SNPs and indels.
- We detailed the impact of IS-related mutations versus other mutations (SNPs and indels) over time.
- As a consequence, we have updated Figures 2b, 3c, 4, and 5b as well as Supplementary Figures 2 and 3 in our revised manuscript.
- Data have been deposited at the Dryad Digital Repository, and the corresponding link is provided.

Please find below point-by-point responses to all of the reviewers' comments.

Reviewer #1 (Remarks to the Author):

The authors present a novel set of analyses of the role of IS mutations during 50,000 generations of experimental evolution of 12 bacterial populations. They show that IS mutations are common, making up to 35% of all mutations in some populations, but that their contribution is two-fold lower in populations with increased point mutation rates (mutators). Further, by exploiting high-resolution genome sequence and fitness data for these populations, they estimate that these IS mutations have on average a small negative effect on fitness, while previous work showed that early IS mutations can also have substantial positive effects. Most remarkably, populations with many IS-related mutations reach lower fitness improvements, suggesting that IS mutations constrain long-term evolvability. The authors also use an elegant assay with a reporter plasmid to estimate rates of IS activity in genotypes from different time points from a population with exceptional IS activity, showing an initial acceleration of IS mutations relative to the increase in number of IS copies, followed by a decline due to mutations outside the IS elements. Together these analyses show that, while IS-mediated mutations may sometimes drive adaptation, they also constrain adaptation in the long term and hypermutability status may help to limit the negative impact of IS mutations by providing alternative, perhaps superior, adaptive variation.

Thank you for this clear and concise summary of our work.

I would like to see this study published in Nat. Comm., but also have a few questions and suggestions, which are hopefully helpful for conveying their message.

Thank you for your help in improving our manuscript.

1. In my opinion, the main advance of this work is the message that IS-mediated mutations, while sometimes adaptive themselves, put constraints on long-term evolvability. But this message was a bit hidden, certainly in the title (which I find little informative), but also in abstract and text. I would

prefer a stronger emphasis on this main message, but I realize that this remains also a matter of taste.

Thank you for these suggestions. We revised the paper's title to make the main findings that you identified as clear as possible: "IS-mediated mutations both promote and constrain evolvability during a long-term experiment with bacteria." We also edited the text in several places to more clearly emphasize the main message, including: the last sentence of the abstract, the last sentence of the Introduction, and the beginning and the last sentence of the Results paragraph on "Contribution of IS elements to fitness during 50,000 generations."

2. In all analyses, except for population Ara+1, no distinction is made between IS insertions and IS-mediated recombination events causing structural variants. It would be instructive to separate these two classes of IS mutations, because one (IS insertions) requires IS activity, while the other does not necessarily require this (recombination may happen between existing copies). It would also be good to define early on what you mean with IS mutations, as it took me some time to realize that both types were included.

We added a statement in the Results section at the beginning of the paragraph titled "Contribution of IS elements to total genomic mutations" in our revised manuscript to define clearly what we mean by IS mutations. Indeed it includes IS transposition events along with rearrangements mediated by recombination between different IS copies on the chromosome.

However, we did not include additional analyses to distinguish between these two types of events because some deletions are associated with an initial transposition event followed by recombination between that new copy and a pre-existing copy of the same IS element. We previously showed that this happened for the deletions affecting the ribose operon (Cooper et al., J. Bacteriol. 2001), and we expect that other recombination events also result from these two successive types of mutational events. Such multi-step changes would interfere with any effort to separately analyze the two types of IS mutations.

3. The regressions used to estimate the fitness effects of IS mutations combine the number of IS mutations with a factor representing time. I found this a bit awkward and expected one or more other factors to explicitly represent the number of other mutations (point mutations, indels). Were such regressions also tried, and if so, what estimates did they yield? Similar estimates for the fitness effects of point mutations and indels would be very helpful for testing whether the reduction of IS mutations in mutator populations was driven superior, or simply more, mutants with point mutations/indels. Also, would it be possible to incorporate a simple form of epistasis, such as diminishing returns epistasis, in these regressions?

We have now performed the analysis in which we also include the number of other mutations (SNPs and indels). The impact of IS mutations is not substantively affected. The impact of SNPs and indels on their own is not strongly related to fitness when controlling for time (generation). Only a marginally significant positive association ($p = 0.03$) was found when considering time as a linear variable. We added a paragraph to the revised manuscript that reports these additional analyses.

Incorporating epistasis per se is not simple. However, we do include the power-law fit for generation time, which emerges from a diminishing-return epistasis model of adaptation.

4. Related to the previous comment, would you have the statistical resolution to ask whether early IS-mutations across all populations (not only Ara+1) might have positive fitness effects, and negative effects only dominate later in evolution? This might help to generalize the adaptive dynamic perspective sketched for Ara+1.

We added a paragraph in the corresponding part of the manuscript in which we examine the impact of IS-related mutations and other mutations (SNP and indels) over time. Indeed, both IS-related and other mutations have a positive impact on fitness over the first 4,000 generations. Subsequently, however, the impact of IS-related mutations on fitness becomes significantly negative, whereas the impact of SNP and indels remains positive (although the magnitude fades and eventually becomes non-significant much later).

5. With respect to the reporter assays to measure the frequency of IS150 mutations in genotypes from population Ara+1, could it be that measured frequencies not only reflect transposition and recombination rates, but also differences in selection? For example, the decline in frequencies after 20,000 generations (Fig. 6b) might also be due to decreased robustness to IS mutations. Perhaps the authors can simply ignore this based on previous work with these lines, but otherwise it might be useful for the readers to discuss this possibility.

As we hypothesize in our manuscript, this decline in the frequency of IS150 mutations after 20,000 generations might be linked to chromosomal mutations that affect the regulation of IS150 activity. However, chromosomal mutations may indeed also affect the cell's robustness to IS mutations. We therefore added sentences raising this possibility in our revised manuscript at the end of the Results paragraph titled "Dynamics of IS-related mutations in Ara+1" and in the Discussion section.

6. On p13, it is speculated that mutations DNA topology, including the *fis* operon and *recF*, may be involved in the early upregulation of IS-150 activity. Were these regions particularly affected by mutations in Ara+1? Similarly, on p16 (bottom) it is mentioned that transposase concentration is involved in regulating IS activity, but it is unclear how transposase is regulated (or rather constitutively expressed).

Mutations in genes regulating DNA supercoiling did occur in population Ara+1, as well as in most of the other LTEE lines. We now provide more details about these mutations in the Results section on "Transposition frequencies of IS150 in Ara+1." However, the specific relevance of these mutations to the burst of IS150 activity in Ara+1 is unclear at this time. Regulation of IS150 involves mostly sliding of the ribosomes to produce a full-length transposase. De-regulation of the translation process might have happened during the period of high IS150 copy number increase. However, this is speculative, and we decided not to add this point in our revised manuscript.

7. In Fig. 4 it is not clear to what time interval these numbers of IS mutations relate (cumulative or only new IS mutations, e.g. 1,000-generation intervals?).

They refer to the number of generations spent by a population in the mutator or non-mutator state and the number of IS-related mutations that fixed during these periods. We clarified this in the figure title, and we added to the legend: "the y-axis values show the number of IS-related mutations that were fixed over the time spent in either the mutator or non-mutator state."

Reviewer #2 (Remarks to the Author):

Review Consuegra et al.

The work by Consuegra et al. is an absolutely outstanding manuscript that provides experimental data on IS evolution that have been theoretically discussed for decades. The statement in the authors' abstract "The long-term dynamics of IS elements and their effects on bacteria are poorly understood" is in my opinion very true. It was until now very unclear how insertion sequences persist in bacterial genomes and what their long term effects are on bacterial genome evolution.

As the authors have noted there has been a long standing debate on what evolutionary forces maintain insertion sequences or other transposons in bacterial genomes. Is it only horizontal transfer between genomes that allows transposons to persist inside the gene pool or do the rare beneficial mutations they cause (one could call an IS element a mutator gene) significantly extend their persistence. To address these questions the authors analyze the insertion sequence content of 12 independent evolution lines of the long term evolution experiment in *E. coli* B over the period of about 60,000 bacterial generations. Their findings are quite extraordinary. Almost every section of the paper adds significantly to our understanding of insertion sequence evolution and its impact on bacterial evolution.

Thank you for these very positive statements.

First, the ancestral IS copy number does not predict IS activity (new insertions) in the experiment. This was completely unexpected to me, but may make sense if large IS families have already been silenced by the host genome (something that could be investigated in future studies). Moreover, different IS elements showed different level of activity (new insertions) across the 12 lines. The authors present evidence that this may be due to historical contingency. If an insertion sequence has caused a beneficial mutation early on, this IS element will be maintained and simultaneously the transposition rate of the IS family will increase due to an increase in IS copy number.

Furthermore, the authors offer several other – maybe even more significant – insights into IS element evolution:

1. IS mutations occurred at twice the rate in non-mutator strains compared to mutator strains.
2. The fitness cost of carrying a single IS element is about 0.6%. Carrying 30 elements reduces your fitness by about 20%. This means carrying IS elements is costly. A basic assumption in most theoretical models, which finally has been experimentally tested at a large scale.
3. In contrast to carrying IS elements being a hypermutator is beneficial.
4. The authors find that there seems to be a negative correlation between the accumulation of IS elements and the evolution of high mutation rates. IS accumulation prevents the evolution of high mutation rates or vice versa. I am quite interested in the causalities of this relationship. As the authors have mentioned, a high mutation rate seems to reduce the transposition rate, but at the same time a high transposition rate reduces the need for a high mutation rate. I think this is a very interesting observation and I can't wait to see this modelled in future studies.

Thank you for so clearly summarizing the main results of our study.

There are a few minor points in the manuscript that the authors could address, discuss in the manuscript or simply reply to in the reviewer response. The manuscript is already full of very interesting data and discussion that it really should be up to the authors whether they want to add

anything else or whether they prefer to leave the manuscript as is.

1. The authors suggest that once IS elements have caused a beneficial mutation the bacterium may be stuck on a suboptimal local adaptive peak. This means at least two mutations are needed to reach a different peak (i.e. first removing the IS elements, which reduces fitness, and then another mutation to reach a different peak). These local optima seem to be quite stable since the authors have not reported any IS element excisions or deletions, and I imagine even if an IS element excises, then the excision is in most cases not scar-less. Could the scar leaving mechanism be a way for an insertion sequence to increase its persistence in the bacterial genome (excision of the IS element is not achieved by the transposition process itself)? How common are IS element deletions (are there genomic deletions that involve IS elements)? Could you estimate the persistence time of IS elements in bacterial genomes based on the available data?

Over the entire dataset, we observed only a handful of IS eliminations, but we have no clear way to estimate precisely the rates of IS retention.

2. Is there evidence of horizontal transfer between strains in a single line? I could imagine that biased gene conversion mediated by the IS element could lead to persistence in the genome, when a subpopulation loses the element. Is it possible that this could be happening within an evolution line?

We do not think there is any horizontal transfer (HT) between cells within the same population. We have two reasons for this view. First, the ancestral strain has no plasmids nor functional prophages, and *E. coli* is not naturally transformable. It is conceivable, however, that some genetic events might have combined remnant, nonfunctional prophages to activate such an element. Second, while the evolution of HT is conceivable (and would be very interesting), in-depth whole-genome analyses find no evidence that it has actually occurred. Several populations have had two lineages coexist for thousands of generations (via crossfeeding or other negative frequency-dependent selection), providing an opportunity to observe HT in the form of homoplasmy at the DNA level—that is, the exact same mutations arising in both lineages *after* they diverged. Despite looking for such events, we do not see any evidence that it has occurred. By contrast, if we introduce conjugative HFR plasmids into LTEE-derived lines, as has been done in another experiment, we readily see extensive horizontal HT leading to genomic recombination (Maddamsetti and Lenski, 2018, *PLoS Genetics*).

3. What would happen if two lines were mixed (one containing lots of IS elements and one without)? The line with lots of IS elements would probably quickly die out. However, it is possible that biased gene conversion could lead to some of the IS elements moving into the genomes lacking IS elements. Has anything like this been attempted? How would IS elements fare if the LTEE environment were structured (i.e. the cultures are mixed every 5000 generations or similar)? Would you see more or fewer strains with an increase in IS element activity?

The Lenski lab has done some experiments (unpublished) in which they have mixed LTEE lines, with the purpose of examining the transitivity of competitive interactions. It is indeed the case that the IS-loaded line, Ara+1, is outcompeted by the other lines, as expected given its lower fitness measured relative to the common ancestor. As explained in the response above, however, HT does not occur in these lines. However, as also noted above, one can design experiments that introduce HT, and it would be interesting to examine the possible persistence of IS elements via the process you describe.

4. Something I found rather confusing is the discussion of the transposition rate per element for IS150. The way I understood the problem, is like the author put it in the last paragraph: “the

increased rate [per element transposition rate] appears to have been driven by the positive feedback between copy number and transposition”. But earlier in the manuscript when the authors measured the transposition rate of IS150 in Ara+1 they argued the opposite point “However, the rates of both processes increased much faster than the number of chromosomal IS150 copies through the 20,000-generation sample (Supplementary Fig. 4), whereas we had expected that the frequency of recombination events, at least, would scale with the number of target copies on the chromosome.”. From my vantage point I can interpret the last sentence in two different ways, both of which are confusing.

First, the authors believe that there should be a linear relationship between transposon copy number and transposition rate. I don't think a linear relationship is plausible (unless I misunderstand something). If we assume a simple reaction norm where a transposon gene produces a transposase enzyme at rate k_1 : $T \rightarrow T+E$ and a transposase enzyme copies a transposase gene at rate k_2 : $T+E \rightarrow 2T+E$ then we get the following differential equations that describe the change in gene number and enzyme number: $dE/dt=k_2*T$ and $dT/dt=k_1*E*T$ if we solve these equations we can calculate the concentration of E and T at different time points: $T(t)=T(0)*\exp(k_1*t*E)$ and $E(t)=k_2*T*t+E(0)$. Hence, we would see an (over)exponential relationship between transposition rate (the number of transposition events measured in the author's experiment) and encoded transposon genes without changing the per element transposition rate k_2 (to explain the data one could for example set k_1 to 1 and k_2 to 1/20 or similar).

Second, I could interpret “scale” to mean that the number of transposon genes scales in an exponential way with the transposition rate. But in this case there is no conflict with the data.

It is possible that I have misunderstand something significant here, but to avoid confusion, I think it would help if the authors could clarify how they believe the number of transposon genes should scale with the transposition rate (i.e. present a simple model that suggests this relationship) and then explain how the experiment does not agree with this model.

We think the answer is based on the two different processes studied. In the first sentence (“the increased rate [per element transposition rate] appears to have been driven by the positive feedback between copy number and transposition”), we refer to long-term dynamics of IS increase through frequency that are in agreement with the formula derived by the reviewer, but which we referred to only qualitatively in the text. In the second statement (“However, the rates of both processes increased much faster than the number of chromosomal IS150 copies through the 20,000-generation sample (Supplementary Fig. 4), whereas we had expected that the frequency of recombination events, at least, would scale with the number of target copies on the chromosome.”), the surprising effect emerged from the recombination rate that scaled up faster than linearly with the number of chromosomal IS150 copies. (The wording “at least” in our previous text may have added confusion, and it has been deleted.) Here, recombination refers to the integration of the plasmid in one of the existing copies independent of the transposase activity, and it should therefore scale linearly.

5. I do not like the data availability statement. I am sure the authors will provide the data/code on demand and that they will make it available after publication. However, I have come across too many papers where it has been incredibly difficult or impossible to obtain data/programs from the author after publication. It would be nice if the authors could make all data available publicly before

publication of the manuscript. I also think this usually makes the paper more accessible and hence may lead to more citations.

Thank you for this suggestion. We have now deposited our data and analysis code to the Dryad Digital Repository, where it is available at the following link: [doi:10.5061/dryad.m37pvmd0v](https://doi.org/10.5061/dryad.m37pvmd0v).

6. Finally, a very minor request. There are some sentences in the introduction, for which citations are missing. It would be nice if the authors could provide those. See for example: lines 31, 41, 54-59 and 69.

We added relevant references at several points in our revised Introduction.

7. line 197/198: “a factor (treating each sampled generation independently);”, what does this mean?

We mean that in the analysis each sampling time point is used as a factor (not as a number), as if it were a name. As a consequence, each time point is associated with a specific mean effect on fitness. We have clarified this point in the text.

Reviewers' Comments:

Reviewer #1:

Remarks to the Author:

I thank the authors for their thorough and detailed response to my comments. I have no further comments and congratulate the authors with a very interesting study.

Arjan de Visser

Reviewer #2:

Remarks to the Author:

I can only reiterate, I think this is an exceptional manuscript and I cannot wait to see it published.

The responses were very thorough and all the misunderstandings are cleared up.

Reviewer #1 (Remarks to the Author):

I thank the authors for their thorough and detailed response to my comments. I have no further comments and congratulate the authors with a very interesting study.

Arjan de Visser

Dear Arjan,

Thank you very much. All your suggestions improved our manuscript.

Reviewer #2 (Remarks to the Author):

I can only reiterate, I think this is an exceptional manuscript and I cannot wait to see it published.

The responses were very thorough and all the misunderstandings are cleared up.

Dear Frederic,

Thank you very much for your helpful work on our manuscript.